# Hybrid LPG-FBG Based High-Resolution Micro Bending Strain Sensor

**DOI:** 10.3390/s21010022

**Published:** 2020-12-22

**Authors:** Song-Bi Lee, Young-Jun Jung, Hun-Kook Choi, Ik-Bu Sohn, Joo-Hyeon Lee

**Affiliations:** 1Department of Cognitive Science, Yonsei University, 50 Yonsei-ro, Seodaemun-gu, Seoul 03722, Korea; 9taniasongbi@yonsei.ac.kr; 2Advanced Photonics Research Institute (APRI), Gwangju Institute of Science and Technology (GIST), 1 Oryong-dong, Buk-gu, Gwangju 500-712, Korea; youngjun.jung@gist.ac.kr (Y.-J.J.); hunkook.choi@gist.ac.kr (H.-K.C.); ibson@gist.ac.kr (I.-B.S.)

**Keywords:** wearable sensor, optical fiber Bragg grating, femtosecond laser, long-period grating

## Abstract

**Simple Summary:**

The solved problem, the helical core sensor for spatial coordinate direction within the fiber was applied to monitor the microscopic repetitive pattern caused by the body structure on the body surface. The body motion state was measured using the strain change of the helical core in three-dimensional space For real-time monitoring, sophisticated sensing and stability against noise are required. The femtosecond laser processing allows us to check the measurement reliability of the sensor through the design of the long and short period gratings. This shows the possibility of monitoring spatial coordinates from one core to one core. It can be used as a wearable sensor.

**Abstract:**

Sensitivity and reliability are essential factors for the practical implementation of a wearable sensor. This study explores the possibility of using a hybrid high-resolution Bragg grating sensor for achieving a fast response to dynamic, continuous motion and Bragg signal pattern monitoring measurement. The wavelength shift pattern for real-time monitoring in picometer units was derived by using femtosecond laser Bragg grating processing on an optical wave path with long-period grating. The possibility of measuring the demodulation system’s Bragg signal pattern on the reflection spectrum of the femtosecond laser precision Bragg process and the long-period grating was confirmed. By demonstrating a practical method of wearing the sensor, the application of wearables was also explored. It is possible to present the applicability of sophisticated micro transformation measurement applications in picometer units.

## 1. Introduction

Recently, the ability to monitor an individual’s activity, health, and lifestyle using sensor technology has arisen. Using wearable “on-body” sensors, various physiological Bragg signals such as heart rate, blood pressure, and respiration rate can be monitored as a person goes about their daily life [1,2,3,4,5,6,7,8,9,10].

Among sensor technologies, the grating Bragg wavelength of fiber optic sensors is very sensitive to environmental variables, such as temperature, stress, bending, and pressure, and thus, it has been widely applied as a high-precision sensor [8]. Fiber optic sensors are used in chemical, bio, and precision measurements for various specific environments. They are being used as hybrid sensors [11,12,13,14]. Fiber Bragg gratings can be applied through grating structures instead of electrical strain gages to obtain useful strain measurements for shape detection algorithms [15,16]. The intensive process using femtosecond lasers can process gratings at strategic locations without removing the fiber’s coating [17].

This study was conducted to investigate the ability of a small, high-precision directional sensitivity sensor with high accuracy to act as a single point sensor through the optical fiber. It is a wearable device that aims to measure body motion.

### 1.1. Wearable Motion Sensing

#### 1.1.1. Body Motion

The body is a complex structure with a high number of degrees of freedom. In a kinematic, multi-joint system, the reference system is established through the chain arrangement system, and the chain must achieve system regulations. The movement trajectory of the point to be measured is patterned to allow the analysis of the spatial and temporal regularity relationship regarding the movement and characteristics of the movement [18,19].

For the segment analysis of a joint in the body system, the position of the sensor in the body segment unit and the number of degrees of freedom in the sensor design must be strategically determined. The vector properties of the joint segment unit and the fundamental 3D surface deformation environment of the body surface require the use of high-precision sensors for dynamic vector motion [20,21].

Human physiological aspects and a dynamic detection monitoring system can be applied when a real-time target feedback system is available. Sensitivity and resolution requirements are required depending on the environment of exceptional contact with the body and sensor. Bragg signal stability and reliability against noise in dynamic environments caused by continuous motion are required. This type of sensor could be used for flexible applications to measure a variety of body vectors. This study involved pattern measurement strain wavelength shift, which could be derived by the femtosecond laser grating process between long-period grating. We detail the possibility of measuring the patterning of a stable Bragg signal on the pico nanoscale using the filtering effect of the grating. Real-time patterned strain measurements, instantaneous moment vector measurements, Bragg signal stability against noise, and ultra-compact practical applications could be used for wearable applications.

#### 1.1.2. Optical Fiber Sensor Sensitivity Improvement for Wearable Motion Sensing

These sensors can be implemented with a double fiber structure, tapered structure, and cladding removal method, which can improve the sensitivity and precision of their external stress and strain response. However, the sensors must be durable and efficient to allow repetitive dynamic movements, and sensors for measuring body applied vectors must be applied to strategic locations in the body’s structural system. Additionally, when using the sensor, it should provide a stable Bragg signal.

##### External Stress and Internal Reaction Applied to Optical Fibers

The wavelength-to-strain response of the Bragg grating is linear. The maximum strain is set to 1% for practical use, and the stress imparted to the fiber by pulling or twisting reacts with only half of the fiber surface of the glass volume. Bending stress has the maximum rate of change in stress for a change in diameter of 10–20 mm due to bending at a constant radius [22,23]. The position of the core within the optical fiber can improve the sensor’s sensitivity.

##### Long-Period Grating (LPG)

Typical LPGs are manufactured to operate at telecommunication wavelengths (1300 nm and 1550 nm). The refractive index of an optical fiber pre-treated with dye immersion in response to a specific wavelength change occurs due to the change in the Si–O–Ge chain structure caused by light irradiation. Coupling of light from core to cladding is induced by periodic modulation of the refractive index along the fiber axis. The loss of guided light in the core at the Bragg wavelength manifests itself as a spectral dip in transmission according to the phase sum of the core mode and the cladding mode, as shown in Equation (1) [22].
(1)$$\lambda_{lpg}=\Lambda \left(n_{eff,cr}-n_{eff,cl}\right)$$
where Λ is the grating period, and $n_{eff,cr}$ and $n_{eff,cl}$ are the useful refractive indices of the core and cladding modes.

The axial strain sensitivity of the LPG may be assessed by differentiating Equation (2) [24].
(2)$$\frac{d\lambda }{d\varepsilon }=\frac{d\lambda }{d(\delta n_{eff)}}\left(\frac{dn_{eff}}{d\varepsilon }-\frac{dn_{cl}}{d\varepsilon }\right)+\Lambda \frac{d\lambda }{d\Lambda }$$

The sensitivity is determined by the material and the waveguide effect. The material effect is the change in fiber dimension, and the waveguide effect is the dispersion term. It occurs in dλ/d𝛬. For an LPG with a periodicity of 100 μm, the material contribution is negative and the waveguide contribution is positive. Choosing to identify the lattice period and fiber composition can create bands with positive, negative, or zero sensitivity to deformation [25].

Depending on the specific measurand or component, there can be different levels of sensitivity and attenuation bands. The waveguide can be designed by changing the configuration of the optical fiber or the LPG spacing [26,27].

Long-period gratings are very sensitive to bending, and strain and refractive index sensors can undergo temperature cross-sensitivity. Moreover, it has been reported that for demodulation systems that detect wavelength shift, a fixed portion of the grating bandwidth is involved. The use of SMF during irradiation of the CO2 laser can provide higher sensitivity [28,29,30,31]. Previous studies manufactured sensors with sensitivity to twisting, bending, and deformation [29]. Due to their advantages, the standard LPFG (SLPFG) interferometer can be used in various applications where information about the bending amplitude and direction are required simultaneously [32]. The SLPFG interferometer of component can have different sensitivity attenuation bands. The waveguide can be designed by changing the configuration of the optical fiber or the LPG spacing.

##### High Precision Integrated Optical Process Using a Femtosecond Laser

Compared to sensors built using conventional UV lasers, femtosecond laser writing gratings can be used for specific applications. The ability to use the grating directly increases the flexibility of the sensor design. It can enable the fabrication of advanced waveguide structures such as multimode enhanced waveguides [33,34,35]. It presents a fast and flexible way to produce superstructure fiber gratings (SFG) using femtosecond laser point-by-point technology. The flexibility to change the grating parameters makes it easy to change the grating length, pitch, and spectral response [36,37].

Point-by-point processing for the optical fiber core is possible, and 3D processing without removing the outer coating is also possible. Femtosecond laser processing allows sensors to be designed for a variety of applications. The overall approach of 3D laser structuring in fiber optics extends to many more applications, including through and blind holes, microfluidic networks, reservoirs, micro-optics and inline Bragg grating waveguides (BGW) filters, birefringence tuning, and interferometers. A fiber Bragg grating (FBG)-based tapered temperature fiber sensor made with a femtosecond laser has been proposed and has shown excellent reproducibility [38,39,40].

### 1.2. Fiber Bragg Grating Sensing

#### 1.2.1. FBG Sensor Principle

The grating Bragg wavelength is widely applied as a high-precision sensor because it is very sensitive to environmental variables such as temperature, stress, bending, and pressure [41,42]. The light reflected from the end of the fiber and the light incident from the standing wave give a periodic energy change inside the fiber core. The grid is created by inducing a regular shift in the refractive index. This phenomenon is called photosensitivity [43,44]. The optical fiber grating device is a photonic device that uses photosensitive properties in which the refractive index changes when a part of the optical fiber core to which germanium or boron is added is exposed to a laser.

##### Fiber Bragg Grating Waveguide Devices

Bragg gratings are widely applied to detect various physical variables, such as temperature, strain, pressure, and acceleration, due to their moderate manufacturing cost and high sensitivity to environmental parameters [45,46,47,48].

Bragg gratings are made by doping the core and irradiating light to adjust the refractive index. They have a sub-micron period and combine the light from the fiber’s forward propagation mode with the reverse propagation mode. They act as a narrowband reflection filter. Interrogators are measured and monitored to identify Bragg center wavelength shift due to changes in the refractive index along the fiber axis of an external tensile load. Strain can be calculated as the moving speed and direction of the external strain [49]. The strain response is expressed as a function of Equation (3) [50].

Various previous studies have aimed to improve the maximum functionality of Bragg grating sensors and approach detection for multiple characteristics of the measurement range [51,52,53,54]. Examples of methods used by previous papers to increase the sensitivity of FBGs are the Bragg grating technique, which uses a fine structure through a taper technique [19,29], and the Bragg grating based on a grating configuration [55,56].
(1 − **Ρ**_e_) **ε** = Δλ/**λ**(3)
(4)$$\frac{\Delta \lambda_{B}}{\lambda_{B}}=\frac{\Delta \left(n_{e}{}_{f\;}\Lambda \right)}{n_{e}{}_{f\;}\Lambda }\;=\;(1+\frac{1}{n_{e}{}_{f\;}}\frac{\partial n_{ef}}{\partial_{\varepsilon }})\Delta \varepsilon =\;(1+{Ρ}_{e})\;\Delta \varepsilon \;=\left(1+{Ρ}_{e}\right)\;\Delta \varepsilon \;=\beta_{e\;}\Delta \varepsilon$$
where Λ is the grating period, $n_{ef}$ is the effective refractive index, λr is the center wavelength of reflection, and Δε is the strain change and $\Delta \lambda_{B}\;$is the strain-induced wavelength shift.
(5)$$\varepsilon =\frac{\lambda_{B}-\lambda_{B0}}{\left(1-{Ρ}_{e}\right)\lambda_{B0}}=\alpha \lambda_{B}+\beta$$
where ${Ρ}_{e}\;$is the photo elastic constant, $\lambda_{B}$-λ$\lambda_{B0}$ is the shift of the Bragg wavelength, $\lambda_{B}$ is the FBG Bragg wavelength without strain, $\lambda_{B0}$ is the FBG Bragg wavelength with strain, and ε is a linear relationship with strain, as shown in Equations (3)–(5) [57].

Previous papers have increased the sensitivity of FBG Bragg gratings using microstructures through tapering technology [58], and Bragg gratings based on grating configurations [59] have been introduced by previous studies. In addition, various previous studies have improved the maximal function of the Bragg grating sensor and investigated the sensing of various characterizations of the measurement range [60,61,62,63].

#### 1.2.2. Interrogator Measurement

The interrogator is measured by the amplitude and phase values of the Bragg wavelength of scattered light along the waveguide. The light emission reaction by deformation is applied to the core by external physical deformation. The change in wavelength as the spacing change function of a fiber Bragg grating is measured as a composite factor of the displacement, the curvature along the axis, the law of physics, and the elastic point, which can be explained by the applied load.

The performance of the interrogator is based on how it measures the center wavelength of the grating. The distribution of all reflected light between the start and end points of the sensor peak is calculated to determine the critical amount of light. The center of the distribution is identified. This includes the specifications of a calculation method suitable for changes in the amount of reflected light distributed.

There are several ways to find the central wavelength of a peak. This method calculates the median by finding the peak’s starting point and the end point of the peak falling below the threshold.

## 2. Materials and Methods

The sensor analyses Bragg signals by measuring the wavelength shift using the significant refractive constant value for the Bragg grating, which relates to strain of the optical fiber axis. In this experiment, to improve the sensor’s strain variation, a single-mode core was used. The fiber core located at the outer 70% of the volume from the center of the fiber. A 1.54 cm pitch helical core was geometrically applied to the external strain as a basis for highly efficient dynamic modeling of the spiral spring element [64,65]. The Bragg grating formed an optical wave path designed as an extended period chirp grating, and the sub-short-period Bragg grating between the long-period grating and the grating was processed with a femtosecond laser. The strain wavelength variation of the sensor Bragg grating was compared between conditions with the presence or absence of the long-period grating.

### 2.1. Optical Fiber

The femtosecond laser process was possible due to the use of germanium-doped to ensure light guiding. Seven helical core structures, one center, and six external positions were used. The specifications were a core spacing of 35 micrometers, a core diameter of 6.3 micrometers, a numerical aperture of 0.21, and a useful core index of within 0.05%.

The preform rotated at high speed to apply twist during the process and had a pitch of 2 cm at a rate of 50 circuits per meter. An optical fiber coated with an acrylate-based UV transparent fiber was used. Figure 1 shows the locations of the seven cores in (a1) and the diameter of the core cross-section in (b). The core sensor used in the Bragg grating is shown in (a1), and in (c), the side of the femtosecond Bragg grating processed core is shown. It is shown as an electron micrograph.

### 2.2. Femtosecond Laser Processing

The pulse repetition rate used was 1 kHz, the output was at least 2 W, the time width was 30 fs(femtosecond)–2 picosecond, and variable conditions were used for the Femtosecond laser—a 785 nm center wavelength, a 185 fs (femtosecond) pulse width, and a maximum output of 1 W (Figure 1a).

The *X* and *Y* axes operated at a maximum speed of 20 mm/s with a precision of 20 nm in the linear motor stage—the 300 mm by 300 mm machining range. The *Z*-axis operated at a maximum speed of 20 mm/s with a precision level of 0.5 m in the ball screw stage and a movement range of 200 mm. The actual operating stage is shown in Figure 2c.

Processing was done by fixing the optical fiber at a delicate precision processing stage and moving the laser beam. The scan speed was adjusted to 2 mm/s, the line spacing was set to 2 μ, and the depth (Z) was fixed at −60 μ. The laser power was changed from 0.3 to 3 mW, and the Bragg grating was processed inside the optical fiber. Figure 2b shows a schematic diagram of the phase mask processing. During processing, the Bragg signal’s detection of the interrogator at 10 kHz occurred in real-time, and the peak generation and shift of the wavelength were observed.

### 2.3. Bragg Grating Specifications of the Sensor

Figure 3 shows the long- and short-period Bragg grid structure diagram of the sensor. The long-period grating (498 μm) (h) and femtosecond laser short-period grating (0.85 μm) (h2) used in UV laser processing were demodulated. The Bragg wavelength passing through the fiber core is measured as the reflected light (e) from the light incident on the fiber (d). The light incident on the long-period grating is divided into light scattered in the cladding mode (f) and light passing through it (g). The summation of the core and cladding modes by the long-period grating can be measured with a spectroscopic spectrometer as opposed to the reflected Bragg grating measurement.

### 2.4. Protocol 1(Spectral Bragg Signal Observation Experiment of Short-Period Gratings Engraved Between Long-Period Gratings)

After connecting the optical fiber of the sensor part and the optical fiber part of the fan-out with a ferrule, a relative rotation position was created to induce light intensity and long-period grating period position modulation. Figure 4 shows the experimental method used.

The relative position of the short-period Bragg grating according to the long-period grating period was measured as the changes in the waveguide angle and length. The fixed point of the passive stage used a rubber stopper fixing frame to prevent the axis of the optical fiber from moving.

The angle unit of the manual stage used was 1 degree, and the length unit was 1 mm. The interrogator used was able to undergo real-time motion monitoring at 10 hz. The device fan-out that connects the 7 cores of the optical fiber to the interrogator individually, one by one, allows the observation of each of the 7 channels. In this experiment, the moving points in Figure 4A(g),B(g) at regular intervals (10 degrees) and length (1 cm) at regular intervals are moved 10 times in Figure 4B, and the wavelength of the interrogator over time observed mutation morphology.

### 2.5. Protocol 2 (Wearable Application Experiment and Sensor Resolution Measurement)

The sensor was placed between the silicone band and the body surface to explore the practical applicability of the wearables. When used directly by humans, the indexing environmental conditions caused by body tremor or temperature were adapted under the manual measuring device. Proceed to the metronome 80′s.

Motion Bragg signal monitoring was performed through mobile phone repetitive numbering. A cellular numbering protocol was performed using the degrees of freedom of the metacarpophalangeal joint centered on the carpometacarpal (CMC) joint [48], as shown in Figure 5c. Rotation at the CMC joint in Figure 5d was generally coupled and posture of the trapezium relative to the third metacarpal changed significantly with thumb position [66,67]. Figure 5e shows the constrictor muscles of thumb movement.

The measuring device and sensor used were the same for both protocols 1 and 2.

## 3. Results

### 3.1. Result 1 (Spectral Bragg Signal Observation Experiment of Short-Period Gratings Engraved Between Long-Period Gratings)

Figure 6A shows a spectrum of the process in which a short-period grating was engraved between the long-period gratings. Figure 6B shows that the sensor part and fan-out part were connected with a ferrule and the seven core positions of the seven core fan-out of the sensor rotated in opposite directions to adjust the intensity of the light power. As the amount of light power increased, the spectra of the long-period grating, and the short-period grating changed in the order shown in B(a–d) of Figure 6.

Figure 6B(e,f) show the three-point wavelength spectrum change according to the long-period spectral period position (Figure 6A(b)) in the state of the highest light power intensity. The case where the short-period grating spectrum was placed in a flat and curved position is shown.

The real-time time-wavelength shift graph in Figure 6B(e) shows a stable level accuracy. The response Bragg signal pattern accuracy of the strain direction wavelength shift to the detected peak point shift improved. The reflection detected threshold was adjustable, and the strain linearity response resulted in a state where the peak loss did not decrease.

The LPG grating plays the role of an ideal bandpass filter. Since the Bragg grating’s Bragg signal measured the peak wavelength shift at the center frequency, it did not cause the Bragg grating’s wavelength shift Bragg signal to attenuate. The reduction in the Bragg grating amplitude did not affect the peak point Bragg signal’s measurement in the mid-band by reducing the bandwidth through a reduction in the lower cutoff frequency.

After connecting the fan-out and the sensor through the ferrule, the outer core was rotated in the direction of a particular angle around the central core, and the optical power through the interrogator and the Bragg’s peak point grating were observed. It was possible to adjust the detected threshold of the reflection without changing the detected peak point. It was possible to confirm the possibility of adjusting the position of the flattening band or the rising or falling band of the peak point through adjusting the optical wave’s position generated by the long-period grating by controlling the intensity of the optical power. Figure 7 shows the spectra of samples with and without long-period gratings.

The amplitude of the FBG reflection spectrum is a function of the LPG spectrum position, and a shift in the wavelength of the LPG spectrum is converted into a change in the amplitude of the FBG reflection spectrum. The external physical strain response can be measured as a shift in wavelength over time. The LPG response of the wavelength shift was observed through experiments in a nonlinear manner. The Bragg grid can be located on a flat or inclined portion of the spectrum. According to the Bragg grating reflection spectrum location, the shift in wavelength response improved with time.

Figure 8 shows the shape of the wavelength shift over time by comparing the cases with long-period gratings. The change in the spatial resolution of the wavelength shift was confirmed.

### 3.2. Result 2 (Wearable Applicational Experiment and Sensor Resolution Measurement)

The search for the possibility of grafting wearable device applications explored the motion repeatability and Bragg signal monitoring patterning in mobile phone numbering. Our primary purpose was to understand the measurement resolution and Bragg signal stability of the Bragg signal, and the characteristics of the hybrid grating structure sensor were applied to the application of wearable devices. While testing the sensor, the body’s body temperature does not affect the sensor’s action.

Figure 9 shows the improvement of Bragg signal precision along with the improvement in spatial resolution shown in Figure 8. Figure 9 shows the detailed morphology comparison of wavelength shift change over time in a 1-s interval. The wavelength shift for B(c) shown in Figure 8 is presented in detail in Figure 9.

The strain direction of the helical core used in this paper with respect to the *X*, *Y*, and *Z* axes of the optical axis and the responsiveness to each axis influenced the sensitivity of the acceleration change. The difference function, the cubic function shown in Figure 9A(b), and the quaternary function shown in Figure 9A(c) were determined. The helical core is parallel to the direction of the waveguide and rotated to the right by 45 degrees. The change of the morphology curve according to the velocity change of the stress strain was observed rather than the general linear waveguide and the longitudinal stress strain response.

The spatial division of the pattern and the pattern Bragg signal’s significance were confirmed within the 1-nanometer range. The linearity of the strain measurement in Pico nano units was verified. Figure 10A shows the morphology of static maintenance at each wavelength shift in the holding state according to a constant stress increase and decrease. Figure 10B shows the morphology of the dynamic wavelength shift in stages with increasing and decreasing angular directions. The dynamic variation was measured according to a continuous 10-degree rise and decrease. C shows the shifts in the long and short wavelength directions repeatedly according to the crossover method in the clockwise and counterclockwise directions.

Figure 11 shows the wavelength shift measured through cell phone numbering. Figure 11a shows the wavelength shift’s mean value according to the mobile phone number. Figure 11b shows the amount of variation in wavelength according to the direction and location of each section. It shows the unit wavelength shift resulting from numbering from left to right or right to left and the change in the diagonal position of the diagonal line from top to bottom or bottom to top. Figure 11c shows the morphology of the wavelength shift over time, according to the numbering used. The flow chart of morphology changed based on the order of the numbering position and direction, and it shows that the position function value was dependent on the relative position before and after numbering. The full set of data is shown in Appendix A and Appendix B.

Figure 12 shows the unit wavelength shift value for each cell phone numbering section, where the accuracy of the delicate repetition pattern is shown to be within 1 nanometer. The accuracy and reliability of the pattern monitoring sensor can be checked in the Pico nano range. The wavelength shift can be expressed as an integer part (N) and an excess fraction part (𝜀).

Figure 13 shows a comparison of the maximum and minimum mean values of short-period Bragg wavelength shifts with and without the use of a long-period grating. A 15 cm flexible loading frame was used to test the Bragg grating sensor epoxy attachment in the middle of the loading frame. Experiment numbers one, two, three, four, five, and six show the comparative values of the maximum, middle, and minimum wavelength shifts in the cell phone numbering experiment. Table 1 shows the sensor reliability results for the phone numbering given in ⒶⒷⒸⒹⒻⒼ in Figure 11. The reliability of the sensor’s effectiveness and the practical application of the sensor’s measurement resolution in pico nano units are shown. From the results in Table 1, the closer to the center point of the short-period FBG wavelength spectrum curve, the maximum sensitivity on the FBG sensing appears to occur.

Figure 10, Figure 12 and Figure 13 show the magnitude of the strain of the external force acting on the axis of the light wave of the sensor when moving the repetitive dynamic trajectory at the speed set to the metronome 80, which is transformed into the Bragg grid by the stress magnitude and direction of the internal stress of the optical fiber is expressed as the shift in wavelength with speed. The degree of correspondence of the periodicity of the waveform generated by the repeated operation means the linearity of the strain, and this was calculated and presented as a reliability.

## 4. Discussion

Compared with conventional sensors, the LPG-based Bragg grating improves the resolution of fiber optic sensors when detecting small parameter changes. Amplitude modulation without reducing the Bragg grating’s power was shown to be possible due to the period of LPG and the coupling factor in the cladding mode, which enabled high resolution to be obtained in the micro bending region.

When the Bragg grating was located in the valley of the long-period grating spectrum, the shape of the wavelength shift was unstable. The reliability and stability of the response morphology iterations were confirmed when on a flat surface than when the response was located in a valley.

The Bragg grating position relative to the long-period grating makes it possible to accurately measure the responsiveness of the distribution response in the fields of chemical, temperature, and biosensors. It can be applied to a sensor that measures the sensitivity distribution of the bias according to the spectrum’s distribution. It is possible to provide a potential factor for improving the measurement sensor’s sensitivity according to the concentration of the bio or chemical sensor.

Development of low-cost sensor devices has been secured by the development of polymer materials for integrated optical devices. The development of polymer materials makes it possible to have excellent thermal stability and chemical resistance of various physical properties. Further studies can be presented with the optical fiber internal stress conditions used in this sensor [68,69,70].

## 5. Conclusions

A high-sensitivity sensor element requires high measurement accuracy in terms of the amount of change in the minority part, and, therefore, it is necessary to control the amount of noise present. The long-period grating can induce a filtering effect with light waves by summing the core and cladding modes, and thus the amount of change in the Bragg signal can be adjusted according to the amount of light. The accuracy of Bragg signal measurement is improved with one accurate frequency measurement without the requirement for continuous monitoring and sensing operations and the Bragg signal is stable in response to noise.

The bending’s sensitivity showed a wavelength shift due to an increase in the moving speed or curvature of the central Bragg wavelength. This curvature can be interpreted as an increase in the segmentation component of the Bragg grating spectrum. The Bragg wavelength’s phase shift can be interpreted as improving the sensitivity to time in the reflection spectrum. It was possible to have detail in measuring the speed change due to the influence of the directional twist strain according to the characteristics of the helical core.

The transmission amplitude modulation of the long-period grating was linked to the reflection of the Bragg grating and the amplitude modulation of the Bragg grating. However, this did not affect the response of the Bragg grid. It was confirmed that the amplitude of the FBG reflection spectrum is a function of the LPG spectrum’s position and it provides a resolution factor for the sensor measurement resolution.

## Figures and Tables

**Figure 1 sensors-21-00022-f001:**
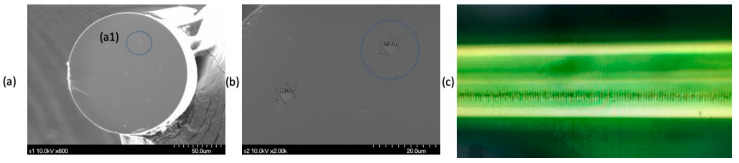
(**a**) cross-section photograph of the optical fiber used. (**b**) a cross-sectional picture of the diameter of the core. (**c**) electron microscope observation photo of the side of the core processed by femtosecond laser grating.

**Figure 2 sensors-21-00022-f002:**
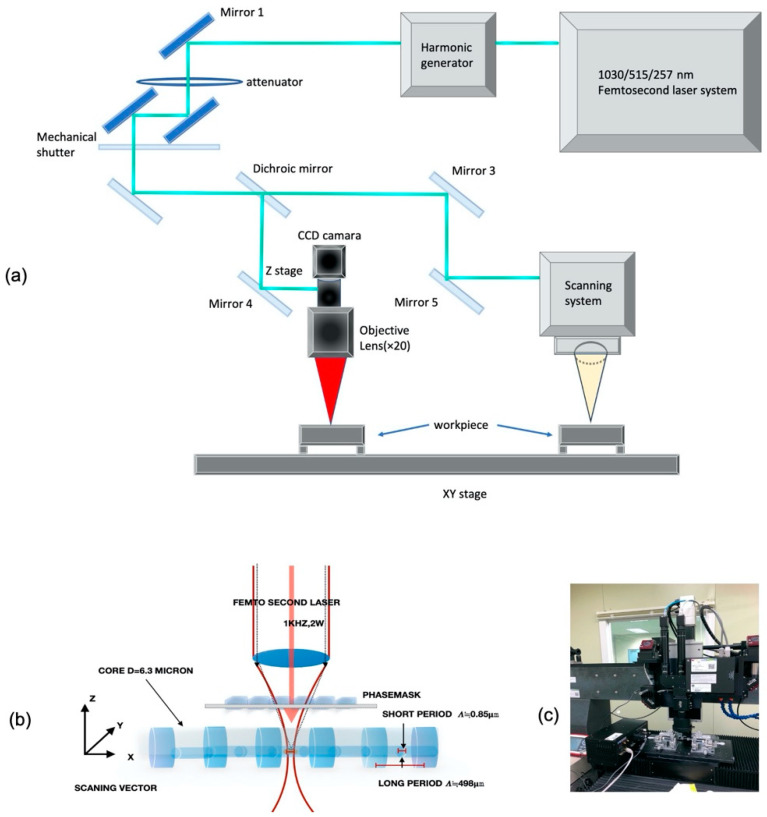
(**a**) schematic diagram of a femtosecond laser micro machining system; (**b**) femtosecond laser phase mask grating process structure diagram; and (**c**) real femtosecond laser precision machining stage.

**Figure 3 sensors-21-00022-f003:**
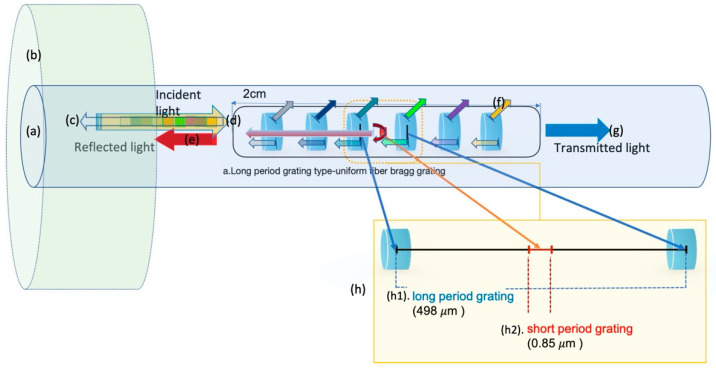
Bragg grating sensor structure and principle. (**a**) fiber optic core; (**b**) fiber optic cladding; (**c**) core reflected light of the long- and short-period Bragg gratings; (**d**) incident light; (**e**) reflected light; (**f**) cladding mode scattering of long-period grating; (**g**) transmitted light; (**h**) long- and short-period grid structure diagram; (**h1**) long-period grating space; and (**h2**) short-period grating space.

**Figure 4 sensors-21-00022-f004:**
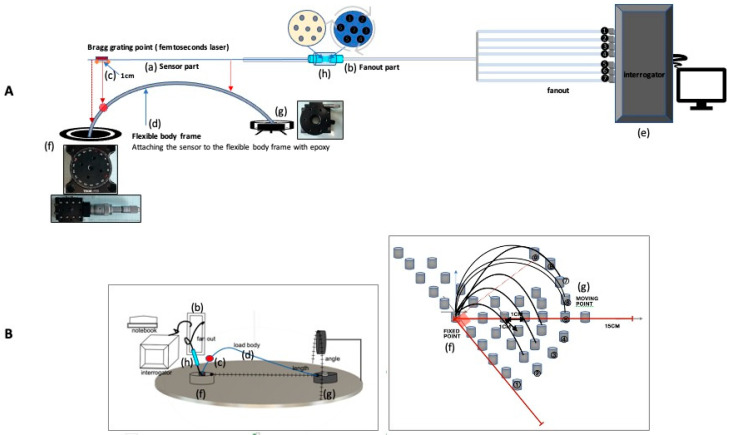
**A**(a–h) Configuration diagram of experimental measurement device; **B**(b–d,f–h) Appearance of the experimental measurement device, diagram of the experimental protocol; **A**(a) manufactured sensor; **A**(b) sensor with the 7-core measurement fanout; **A**(c) short-period Bragg grating position; **A**(d) flexible loading body frame; **A**(e) interrogator and notebook; **A**(f) and **B**(f) center fixed point manual stage of the sensor movement; **A**(g) and **B**(g) strain deformation measurement the moving point and manual measuring stage; **A**(h) schematic of the ferrule connection between the sensor part and fan-out measurement part; **B**. Configuration diagram and experiment method sequence diagram of the actual test equipment.

**Figure 5 sensors-21-00022-f005:**
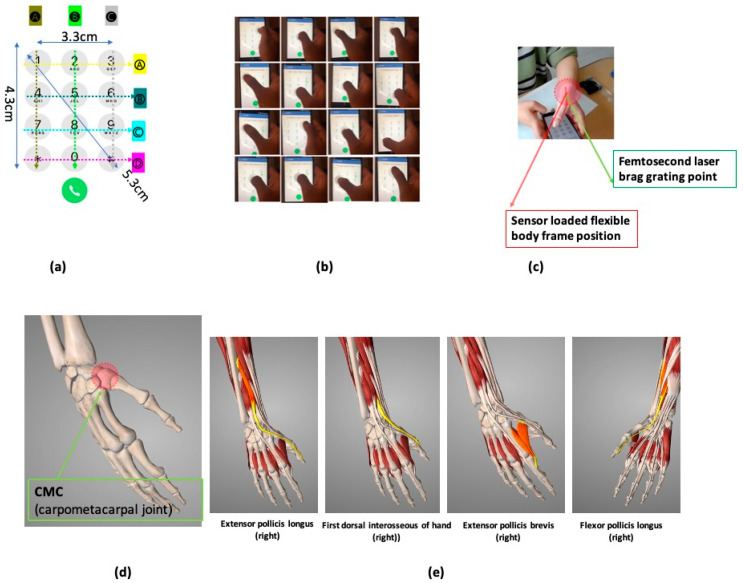
(**a**) mobile phone numbering coordinate standard; (**b**) real photo of the cell phone numbering experiment; and (**c**) sensor-wearing method and experiment method photo. (**d**) carpometacarpal (CMC) joint (**e**) constrictor muscle combined with rotation of the CMC joint.

**Figure 6 sensors-21-00022-f006:**
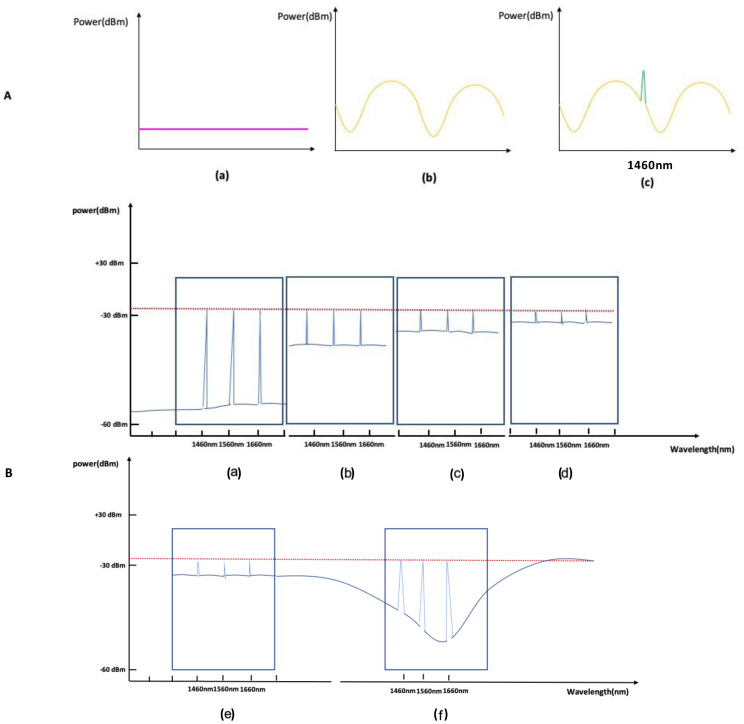
(**A**). (a) interrogator measurement spectrum before the grating process; (b) interrogator measurement spectrum after the long-period grating process on the entire optical fiber; and (c) grating spectrum of one point of the short-period grating between long-period gratings with a femtosecond laser. (**B**). (a) sensor Bragg wavelength spectrum with minimal light power intensity; (b) sensor Bragg wavelength spectrum with 30% light power intensity; (c) sensor Bragg wavelength spectrum with 70% light power intensity; (d) sensor Bragg wavelength spectrum at the maximum light power intensity; (e) three short-period Bragg wavelength spectra placed flat period part of the long-period grating; and (f) three short-period Bragg wavelength spectra placed on curved part of the long-period grating.

**Figure 7 sensors-21-00022-f007:**
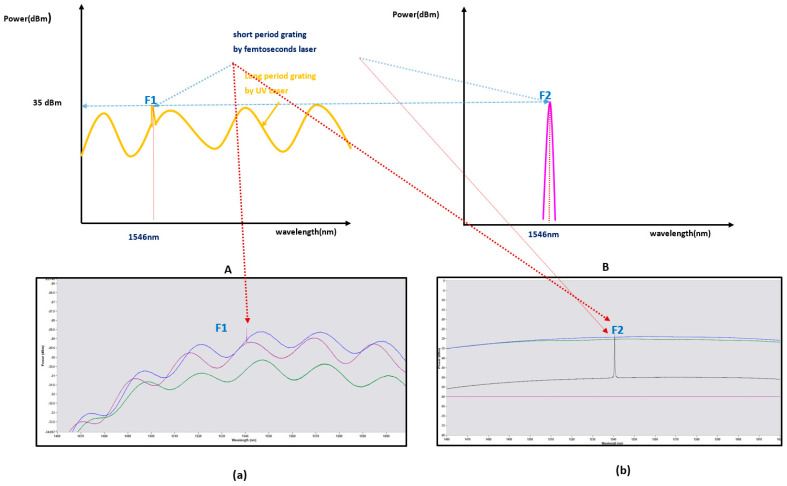
(**A**) One short-period Bragg grating spectrum between long-period gratings; (a) **A**’s morphology picture measured on an actual interrogator monitor; (**B**). Bragg grating spectrum with only short-period gratings; (b) **B**’s morphology picture measured on an actual interrogator monitor.

**Figure 8 sensors-21-00022-f008:**
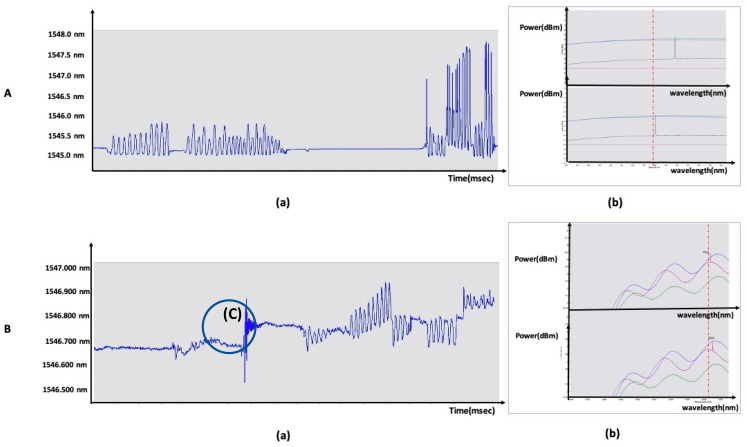
(**A**) (a), (b) time-dependent strain wavelength displacement morphology and spectrum of short-period grating process sensors; (**B**) (a), (b) time-dependent strain wavelength displacement morphology and spectrum of short-period Bragg grating process sensors between long-period gratings.

**Figure 9 sensors-21-00022-f009:**
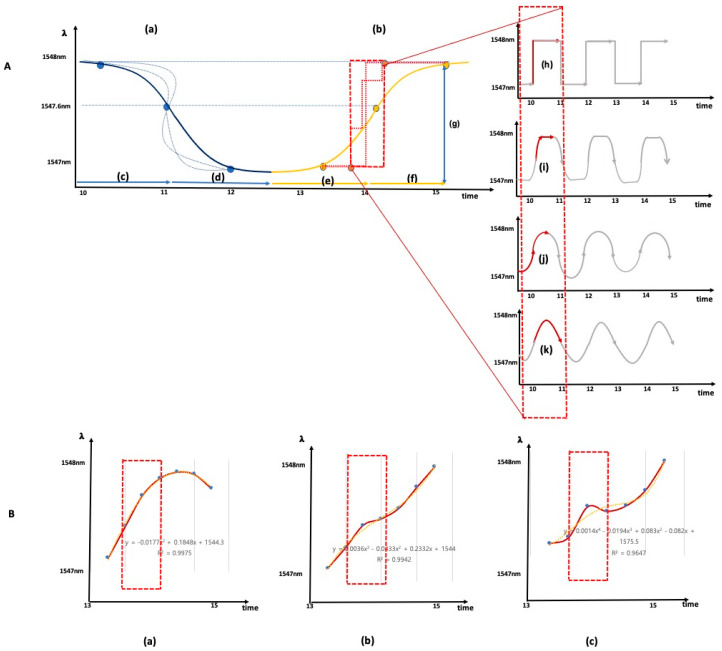
(**A**) (a–k) wavelength variation according to strain deformation, (**B**) (a–c) wavelength variation according to strain velocity variation (**;A**) (a) example of curve deformation according to continuous the speed increase and decrease in the strain application; (**A**) (b) example of stepped deformation according to the continuous presence or absence of strain application; (**A**) (c,e) morphology when increasing the strain application rate; (**A**) (d,f) morphology when reducing the strain application speed; and (**A**) (h–k) morphology according to changes in the acceleration of the strain application. The three-step morphology based on the change in the magnitude of the stress applied by acceleration is shown as ((**B**) (a) < (**B**) (b) < (**B**) (c)).

**Figure 10 sensors-21-00022-f010:**
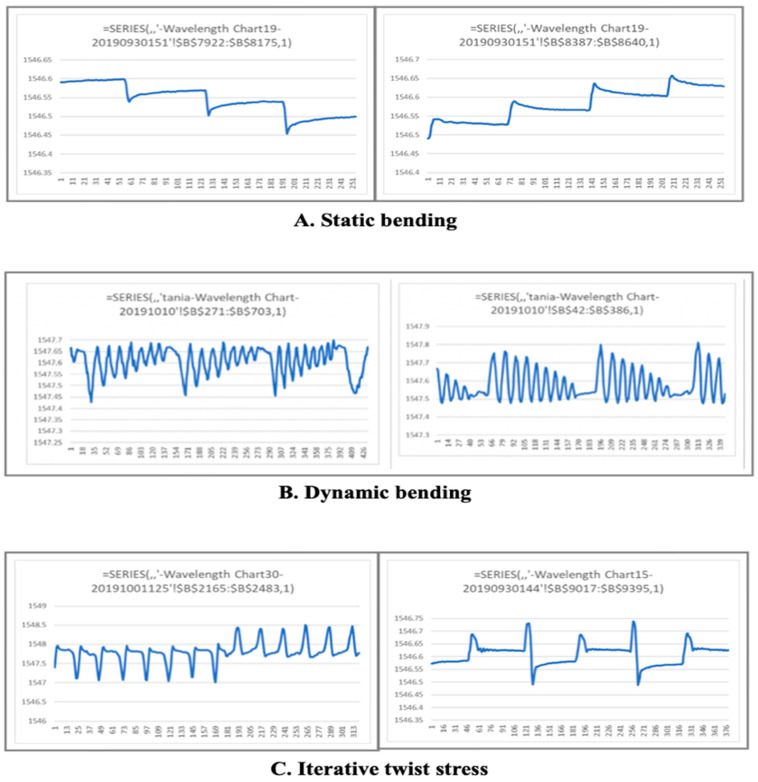
(**A**). The morphology of the wavelength shift when strain decreases and increases in static steps (picosecond); (**B**). Continuous stepwise strain reduction and increase in the wavelength shift morphology; and (**C**). Wavelength shift pattern morphology of clockwise and counterclockwise directional stress–strain.

**Figure 11 sensors-21-00022-f011:**
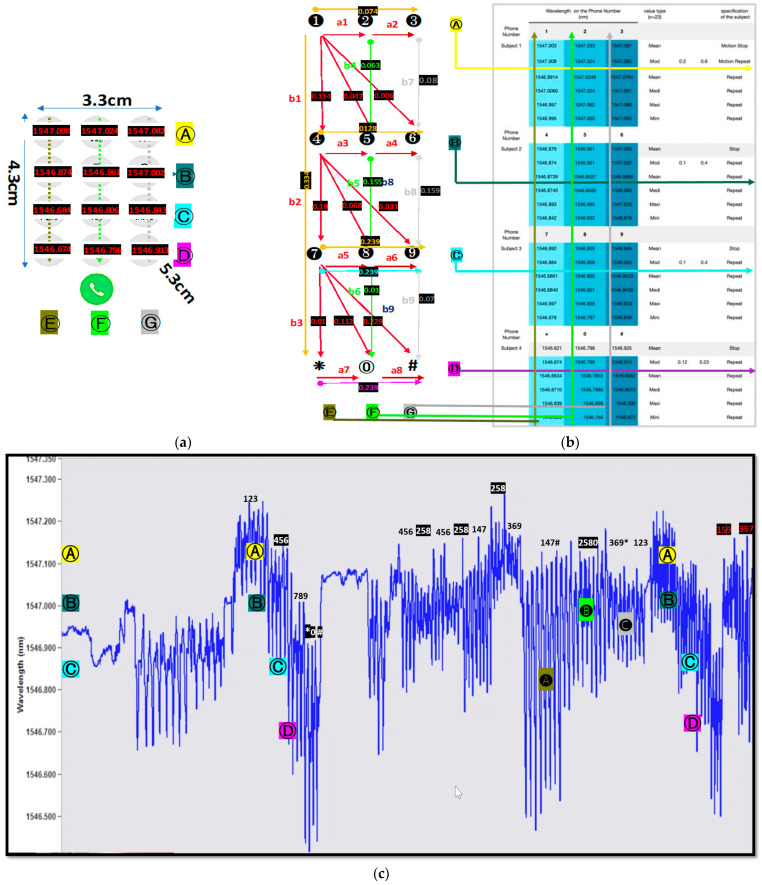
(**a**) numbering definition Ⓐ123 Ⓑ456 Ⓒ789 Ⓓ*0# Ⓔ147* Ⓕ2580 Ⓖ369# and experiment result value (**b**) direction of thumb movement and experiment result value (**c**) real-time numbering wavelength shift morphology.

**Figure 12 sensors-21-00022-f012:**
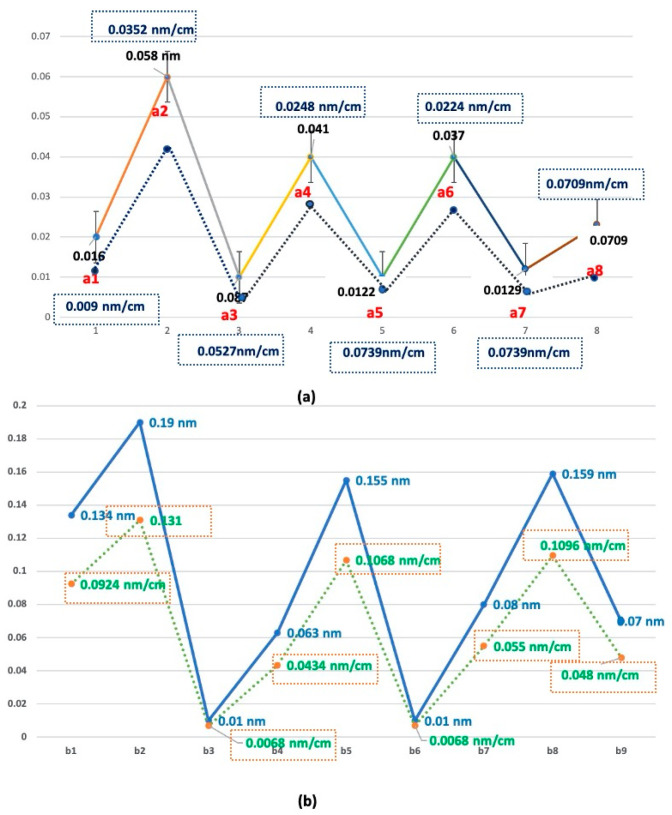
(**a**) unit value of the wavelength shift in horizontal section a and (**b**) unit value of the wavelength shift in vertical section b.

**Figure 13 sensors-21-00022-f013:**
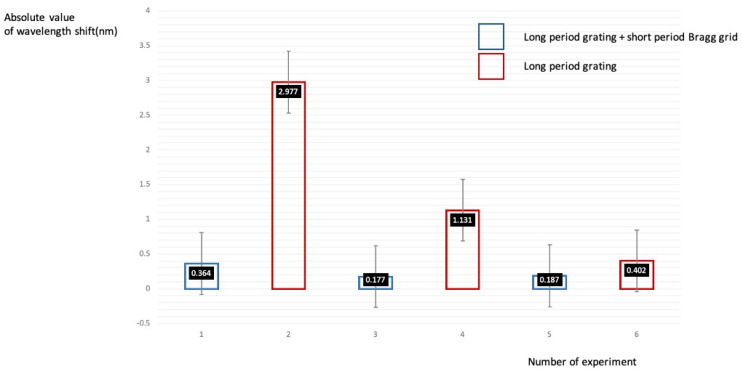
Comparison of the Bragg wavelength shift with and without the use of a long-period grating.

**Table 1 sensors-21-00022-t001:** Sensor reliability results for repetitive motion.

Experiment Number	Standardized Cronbach’s Alpha(α)	Difference
1	0.894	−0.046
2	0.852	−0.035
3	0.861	0.018
4	0.92	−0.026
5	0.887	−0.028
6	0.885	−0.037
7	0.88	0.042
**Standardized** **Cronbach’s alpha**	0.894	
	**α** ≥ 0.9(excellent),	
	0.8 ≤ **α** ≤ 0.9(good),	
	0.7 ≤ **α** ≤ 0.8(acceptable)	
	0.6 ≤ **α** ≤ 0.7(questionable),	
	0.5 ≤ **α** ≤ 0.6(poor),	
	**α** ≤ 0.5(unacceptable)	

## Data Availability

Data sharing not applicable.

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
