# Peer review of "Hybrid LPG-FBG Based High-Resolution Micro Bending Strain Sensor"

_sensors, 2020, doi:10.3390/s21010022_

Round 1

Reviewer 1 Report

A technical sound experimental setup.

The manuscript can be improved in areas such as

  1. Providing more explanations on interrogator reading the peaks, effect of manual measuring stage and femtosecond laser writing the grating consistency (and yield) on the result
  2. Suggest to present the figures and table more organised and clearly such as labeling the axis and using professional software to plot the graphs instead of ENLIGHT
  3. A better way to name the labels such as a,b,c...z naming instead of c1,c2,c3
  4. Check headings, formatting, upper/lower case and styles to be inline with publisher standard
  5. Grammar, adverbs, tenses

Reviewer 2 Report

Comments to the manuscript entitled “Hybrid LPG-FBG based high resolution micro banding strain sensor“

The manuscript's content lacks scientific language, references, and clear explanations and correlations regarding the investigated theme. The combination of the lack of these components gives the reader a terrible reading experience. The first impression, which is extremely important in any evaluation, is that this manuscript is a poorly written technical manuscript.

Due to the investigated theme's attractiveness, I have decided to benefit from the authors' doubt and did a second reading of the manuscript disregarding the non-scientific style and language. By focusing only on the results, I have concluded that they are scientific meaningful; therefore, this manuscript can be published in this respectful journal after a MAJOR REVISION process, particularly regarding the scientific soundness of the manuscript.

Among the several issues present in the manuscript, I will give just a few examples, but the authors should revise the entire text to achieve a final manuscript with scientific appeal and soundness:

(Section “Introduction”) As the authors mentioned, “sensor technology has reached an unprecedented level”, meaning that several examples of such technology are already in the market, or in the worst case, there are massive publications. The reader needs more than six references to understand the potential of such research field;

(Section “Introduction”) The authors mentioned: “This study was carried out to monitor a more sophisticated and faster response than conventional sensor elements”. Define “conventional sensor elements” with the appropriate references. Also, define “various body vectors”.

(Section “Introduction”) The authors should drastically improve this section.

(Section “1.1 LPG” – Line 51) Insert the wavelength units.

(Section “1.1 LPG”) Equation 1: needs a reference; Equation 2: Epsilon was not defined. Its definition only appears in Equation 3.

(Section “1.1 LPG”) No connection between the paragraph ends on line 66 and starts on line 77.

(Section “1.2) What does mean a “modest fabrication”?!

(Section “1.3”) Please define “conventional UV laser”; Compare in qualitatively terms the differences between microfabrication results using a LASER in different temporal regimes, such as nanosecond (ns), picosecond (ps), and femtosecond (fs); Explain to the reader, scientifically, the advantages of an fs micro fabrication, in particular, in the micro grating fabrication.

(Section “2.2.”) Re-write the entire sentence!

(Section “3”) Besides the language editing, the results, as I have mentioned before, seem meaningful, although their discussion should be improved to avoid only technical soundness. Some figures should be re-done once the Y- and X-axis caption are not legible. A scientific manuscript should also have the same pattern figures, i.e., the authors should use the same analysis and graph software to create all the figures.

(Section “4” and “5) Both sections could be merged; Another incoherence is that the authors emphasize using an fs LASER to produce the gratings, but they did not mention this feature in these sections! Why?

Summarizing my review:

(1)        As I have mentioned, this manuscript needs MAJOR REVISIONS to sound like a scientific manuscript. I will not accept this manuscript for publishing without such MAJOR REVISIONS, which go further than the few examples I listed before.

(2)        After such a revision process, I will be able to review in detail the results, which seem OK. Still, I prefer to give the final review regarding the results and discussion after the authors' first revision process.

Reviewer 3 Report

This paper describes the design and test of a bending wearable sensor to be implemented on body for shape/movement sensing (haptics applications). With this aim in view, Fiber Bragg Gratings (FBG) and Long-Period Gratings (LPG) are femtosecond photowritten in a single core of a multicore fiber composed of a core placed at the fiber center and 6 other cores placed at 60° from each other along a radius of approximately 50% the outside diameter.

The English is poor and must be improved, it makes the reading difficult. This paper must be proof-read by native English people.

Furthermore, numerous misinterpretations and mistakes hamper the understanding. For instance, the term "banding" should be read "bending" ("twisting"). The term "Bragging" signal is improper (Bragg signal is acceptable). Many terms are not explained (SFG, BGW ?). Units are often missing (e.g. a 1.54 pitch helical core, 1.54 what ?). Several contradictions are also apparent, for instance in the discussion part (when the FBG is located in the valley … the response is flat, … but when it is located on a valley, the response is not flat).

Even more important is the lack of useful technical description that renders the paper largely incomprehensible. The experimental devices are not described (Bragg interrogator, fan-out part). Bragg wavelength changes are not linked with operational conditions so that the reader is unable to figure out how the sensor behaves in practice. The global paper mostly shows qualitative data with few quantitative results that are also unclear (what does stress strain wavelength shift morphology ? mean, what is measured : stress or strain ?, or something else ?).

The authors use a LPG as spectral filtering element, modifying the amplitude of the FBGs (short-period gratings). One would expect that the LPG would have been used as wavelength-to-amplitude converter (edge filter method). However, the Bragg spectra are monitored with a multichannel Bragg interrogator and the useful signals are the FBG wavelengths. So the question is simple : what is the LPG used for ?

Furthermore, most shape sensors involve at least 3 FBGs photowritten over 3 cores placed at 120° from each other. Both the radius of curvature and angular orientation are then retrieved from a combination of strain data experienced by each core.

In this paper, only one core is photowritten and no other FBGs are photowritten on the 6 remaining cores. So other questions arise, why using a 7-core fiber ? Why only one core was actually used for sensing ? Is the fiber rotated along its axis so as to provide the maximum bending sensitivity vs angle ? Is the LPG photowritten on the same core as the FBGs ?

The paper also lacks a state of the art in bending measurement with FBGs and multicore fibers, for instance, the papers of H. Zhang, J. Opt. 18, 2016, pp. 085705 or F. Khan, IEEE Sensors J., 19 (14), 2019, pp. 5878-5884.

A state-of-the-art of FBGs for sensing is also necessary because the descriptions of photosensitivity and metrological properties (parts 1.1 and 1.2) are not clear. I suggest to add review references such as A. Othonos, Rev. Sci. Instrum., 68 (12), 1997, pp. 4309-4341, or Y-J. Rao, Meas. Sci. Technol., 8 (4), 1997, pp. 355-375.

A minimum state-of-the art is also necessary for LPGs, for instance the papers of A.M. Vengsarkar, J. Lightwave Technol. 14 (1), 1996, pp. 58-65 and V. Bhatia, Opt. Lett., 21 (9), 1996, pp. 692-694.

I cannot get Eq. (2) by differentiating Eq. (1). If Eq. (2) is a new result, then the mathematical demonstration should be given. If Eq. (2) is already demonstrated by previous authors, then please provide adequate references.

Reviewer 4 Report

The paper presents a study about the use of an hybrid technology combining an LPG-FBG with the goal of high resolution micro banding strain sensor. I have some comments that need to be addressed to improve the quality of paper and some doubts.

1) In the introduction I think it is can be improved and well written and I miss some recent works about wearable and sensing in general. Please include: a) Fiber Bragg Gratings for medical applications and future challenges: A review, IEEE Access 8, 156863-156888, 2020. b) Polymer optical fiber-based sensor for simultaneous measurement of breath and heart rate under dynamic movements, Optics & Laser Technology 109, 429-436, 2019; c) Smart textiles for multimodal wearable sensing using highly stretchable multiplexed optical fiber system, Scientific Reports 10 (1), 1-12, 2020. 

2) In addition to Introduction, when the authors claim: "Fiber optic sensors are used as hybrid sensors for various particular environments in chemical, bio, and precision measurements" - It is weak number of reference and impact. Please consider: d) Cortisol in-fiber ultrasensitive plasmonic immunosensing, IEEE Sensors Journal, 2020; e) HER2 biosensing through SPR-envelope tracking in plasmonic optical fiber gratings, Biomedical Optics Express 11 (9), 4862-4871, 2019.

3) Figures quality must be improved.

4) How about the temperature influence on the sensors when is in hand, like figure 5? Please add details.

5) I miss a table with comparison with literature containing the main key parameters that bring novelty and impact to this work compared with the published ones.

6) The fabrication of such structure using fs laser, etc, is a complex and expensive system. How about the state-of-art to get other configuration like using MZI ou cavities to get similar performance? A fs laser could be operate without phase mask method if good translation stages have. For such work, with a low cost UV laser system, you could inscribe gratings (FBG and LPG) in a relative low cost manner. Please consider to read and add some words about the following references: Inscription of Bragg gratings in undoped PMMA mPOF with Nd: YAG laser at 266 nm wavelength, Optics Express 27 (26), 38039-38048, 2019; Fast inscription of long period gratings in microstructured polymer optical fibers, IEEE Sensors Journal 18 (5), 1919-1923, 2018.

7) It is mandatory to be highlight in abstract and conclusion and even somewhere in the text about the "high resolution micro banding strain sensor" as reported in the title. Is not clear in the document and comparing with literature.

8) many typos and errors along the text. Fig. 5 c) is not Brag but Bragg, among others.

Round 2

Reviewer 1 Report

Significant improvements and effort have been made to the manuscript.

The authors have used correction service. A native English edits will greatly improve the quality of the paper.

To improve readability of information such as but not limited to the followings:
- line 61 and 117 on repeated headings "1.2"
- line 251, 256 and 257 on CMC definition in the text first
- line 279-281 seems to describe the opposite from figure 6B
- line 79 and 80 on upper/lower case "Cr" vs "cr"
- line 106, 192 on punctuation "."
- figure 11 is hard to read and identify the small/blur wordings
- line 592 reference has unnecessary author information

Author Response

Please check the file attachment. & we received English proofreading from mdpi.

Reviewer 2 Report

I want to thank the authors for the manuscript revision.

Author Response

(The authors gave the same response as above.)

Reviewer 3 Report

The authors revised their paper significantly and its purpose and methods are now better defined. However, it still contains errors and unclear statements. Furthermore, the English is still poor and needs considerable improvements.

In the title "banding" should be changed by "bending".

Throughout the paper, the authors use the dual term "stress-strain" although the Bragg shift only relates to strain. So please, remove "stress" anywhere mentioned because this parameter is not relevant.

In line 69, the sentence "one percent of the external strain has a linear dependence on the grating" is improper. I suggest writing: "the wavelength-to-strain response of the Bragg grating is linear. The maximum strain is set to 1 % for practical use".

In line 78, the sentence "Periodic modulation by irradiation of light forms a spectrum" is improper as well. I suggest writing: "Coupling of light from core to cladding is induced by periodic modulation of the refractive index along the fiber axis. The loss of guided light in the core at the Bragg wavelength manifests itself as a spectral dip in transmission".

In line 82, the reference that pertains to Eq. (2) is [47] (Ralph Tatam et al.), not [46].

In line 143, Δε is the strain change (with respect to a reference state) and ΔλB is the strain-induced wavelength shift.

The paragraph from line 153 to 157 is incomprehensible and must be rewritten. Bragg sensing mostly involves spectrometric measurements (i.e. recording of spectra). Idem for the paragraph from line 244 to 247.

In line 170, what does "micro-direction measurement monitoring" mean ?

In line 176, the core of most telecommunication-grade fibers is germanium-doped to ensure light guiding. It is not a fiber "treatment".

In line 264, change perrule by ferrule.

In line 323, what does "morphology of the resolution" mean ?

In Figure 13: How do you account for the improvement in sensitivity due the use of the LPG ? The maximum sensitivity seems to occur for FBG wavelengths located on the edge of the LPG spectral curve.

Author Response

(The authors gave the same response as above.)

Reviewer 4 Report

The paper can be published after the authors addressed all my comments in a good way.

Author Response

(The authors gave the same response as above.)
